# Differentiating Lumbar Spinal Etiology from Peripheral Plexopathies

**DOI:** 10.3390/biomedicines11030756

**Published:** 2023-03-02

**Authors:** Marco Foreman, Krisna Maddy, Aashay Patel, Akshay Reddy, Meredith Costello, Brandon Lucke-Wold

**Affiliations:** 1Department of Neurosurgery, University of Florida, Gainesville, FL 32610, USA; 2Department of Neurosurgery, University of Miami, Miami, FL 33136, USA

**Keywords:** lumbar spine pathology, sciatica, peripheral plexopathy, lumbosacral plexopathy, regenerative nerve therapy

## Abstract

Clinicians have managed and treated lower back pain since the earliest days of practice. Historically, lower back pain and its accompanying symptoms of radiating leg pain and muscle weakness have been recognized to be due to any of the various lumbar spine pathologies that lead to the compression of the lumbar nerves at the root, the most common of which is the radiculopathy known as sciatica. More recently, however, with the increased rise in chronic diseases, the importance of differentially diagnosing a similarly presenting pathology, known as lumbosacral plexopathy, cannot be understated. Given the similar clinical presentation of lumbar spine pathologies and lumbosacral plexopathies, it can be difficult to differentiate these two diagnoses in the clinical setting. Resultingly, the inappropriate diagnosis of either pathology can result in ineffective clinical management. Thus, this review aims to aid in the clinical differentiation between lumbar spine pathology and lumbosacral plexopathy. Specifically, this paper delves into spine and plexus anatomy, delineates the clinical assessment of both pathologies, and highlights powerful diagnostic tools in the hopes of bolstering appropriate diagnosis and treatment. Lastly, this review will describe emerging treatment options for both pathologies in the preclinical and clinical realms, with a special emphasis on regenerative nerve therapies.

## 1. Introduction

In the most recent epidemiological overview on lower back pain (LBP), it was found that up to 80% of individuals worldwide will present with an episode of LBP throughout their lifetime [1]. More specifically, LBP can present as pain, muscle tension, or stiffness localized below the costal margin and above the inferior gluteal sulcus, with or without associated leg or foot pain [2]. Regarding the latter major symptom of LBP, known as sciatica, this is the most common sequelae of lumbar spinal pathologies and is reported to affect as much as 40% of the adult population at some time [3]. Clinically, sciatica is diagnosed due to its characteristic presentation of radiating pain and is the principal radiculopathy affecting the lumbar spine [4].

A separate yet related set of more rare disorders are known as peripheral plexopathies. In this paper, we focus on disorders involving the lumbosacral plexus and its tributaries such as the femoral and sciatic nerves. In contrast to radiculopathies such as sciatica, which commonly only affect a single nerve root, plexopathies are clinically defined as the involvement of at least two different root levels and from at least two additional peripheral nerves [5]. As such, the clinical signs and symptoms of plexopathy—weakness, loss of tendon reflexes, sensory deficits, and often pain—tend to follow a dermatomal distribution, localized to the involved neural structures providing peripheral nervous system innervation to the afflicted muscle groups [6,7].

However, differentiating lumbar spinal pathologies from peripheral plexopathies involving the lumbosacral plexus has historically proven to be clinically challenging. Particularly, this fact holds true because neurologic findings are not always well defined as described above and because there is considerable overlap in the sensory and motor territories supplied by the individual lumbar nerve roots, with the lumbosacral plexus or the nerves derived from the lumbosacral plexus [6,8]. Thus, as two separate pathologic entities that affect the population with opposing levels of burden, it is crucial to elucidate their differentiation to increase diagnostic efficacy and appropriate treatment.

## 2. Anatomy of the Lumbar Spine and Lumbosacral Plexus

A comprehensive understanding of the functional anatomy of the lumbar spine and the various plexuses that supply the lower limbs and pelvic girdle is warranted when discussing their respective etiology and differentiation in a clinical setting.

The lumbar spine consists of five bones in the lower back, known as the L1–L5 vertebrae, which are composed of two parts, the vertebral body and the vertebral (neural) arch [9]. Further, each of these vertebrae have numerous osseous structures, as seen in Figure 1, and articulate to form distinct anatomical structures such as the intervertebral foramen, which transmit the lumbar spinal nerves and the associated radicular arteries that supply the spinal cord [10]. The lumbar spine is segmented into its five component vertebrae by structures known as intervertebral discs, with the vertebral bodies increasing in size as the column descends [11]. These discs are composed of a centrally located nucleus pulposus concentrically encircled by the annulus fibrosis and a cartilaginous endplate that forms the interface between the intervertebral disc and the adjacent vertebrae [12]. Together, these fibro-cartilaginous structures impart the lumbar spine with multiples ranges of motion—including flexion, extension, lateral bending and rotation—as well as the ability to transfer and distribute spinal loads [13].

On the whole, the lumbar spine is responsible for several functions including upper body support and weight distribution, protection of the spinal cord/cauda equina, and—by extension of the nerves that branch off of the lower spinal cord and cauda equina—leg sensation and movement [14]. Regarding the latter functions, it is important to note that there are five pairs of corresponding lumbar spinal nerves (L1–L5) that innervate the lower limbs. Moreover, as seen in Figure 2, these innervations generally follow skin- and muscle-specific distributions supplied by the dorsal and ventral root fibers of a given spinal nerve, known as dermatomes and myotomes, respectively [14]. Specifically, the L1 spinal nerve provides sensation to the groin and genital region as well as motor innervation to the hip; the L2 spinal nerve similarly provides motor innervation to muscles of the hip for movement such as abduction; together, spinal nerves L2–L4 give sensation to the anterior aspect of the thigh and the medial aspect of the lower leg, with spinal nerves L3 and L4 providing motor innervation for movements such as knee extension; finally, spinal nerve L5 provides sensation to the lateral aspect of the lower leg, the dorsum of the foot and the space between the first and second toe, as well as corresponding motor innervation for movements including knee flexion and great toe extension [14,15,16,17].

Additionally, it is pertinent to underscore that once the lumbar spinal nerves exit the intervertebral foramen, they become a related yet clinically distinct set of anatomical structures known as the lumbar and sacral plexuses—which are together known as the lumbosacral plexus (LSP). The LSP is a complex peripheral nervous system structure that similarly supplies sensory and motor innervation to the lower limbs and is derived from the anterior rami of the L1-S4 nerve roots, and a small contribution from T12 [5]. As illustrated in Figure 3, the LPS has many component nerves and penetrates the psoas major muscle, later emerging on the lateral aspect of the pelvis [18].

Summarized in Table 1, the LSP has an inherently complex set of dermatomal and myotomal innervations and serves as the extension to the spinal nerves with which they are derived from. As a result, at the clinical level, insult to any given component(s) of the LSP can have overlapping sensory and/or motor deficits as those seen in lumbar spinal nerve etiology. In this paper, we will identify common pathologies of the lumbar spine and the lumbosacral plexus, as well as compare their presentation, evaluation, treatment, and management.

## 3. Etiology

Upon extensive review of the literature, the most common etiology of sciatica due to lumbar spinal pathology is caused by lumbar spinal stenosis. As seen in Figure 4 below, however, the pathogenesis that underlie said stenosis cannot be simply traced back to one causative agent, but a multiplicity of pathologies. Thus, the most common mechanisms that lead to the development of lumbar spinal stenosis in the general population will be detailed in this section. Similarly, LS plexopathies also have a complex set of etiologies and thus the most prevalent and clinically significant causes will be discussed.

### 3.1. Lumbar Spine Pathology

#### 3.1.1. Herniated Nucleus Pulposus

Lumbar disc herniation occurs when the annulus fibrosus ruptures leading to the nucleus pulposus protruding from the intervertebral disc space, also known as herniated nucleus pulposus (HNP) [21]. HNP is one of the most common lumbar spinal pathologies, with 95% of cases occurring at the L4-L5 or L5-S1 level and is one of the main contributors to lumbar spinal stenosis [22,23]. Peak incidence is between ages 30–50, with the prevalence doubled in males [24]. Because of the thicker anterior longitudinal ligament, herniation is most likely to occur in the posterior direction—more specifically the posterolateral direction—leading to compression of the nerve root. With each nerve root having a specific motor and sensory area of innervation, pain and/or paresthesia corresponding with the effected nerve root follow the same distribution [25]. Coughing, sneezing, or straining may worsen symptoms by increasing pressure in the surrounding areas.

#### 3.1.2. Osteoarthritis

A second major contributor to lumbar spine stenosis can be found in the degenerative process known as osteoarthritis. Although arthritis can affect any joint, weight-bearing joints are most susceptible to arthritic changes including the spinal column [26]. The spinal column is composed of three joints: one intervertebral disc and two facet joints, which are all susceptible to degenerative changes [27,28,29]. This degeneration leads to disc space narrowing, the formation of osteophytes, facet joint osteoarthritis, and eventually foraminal narrowing [29,30]. Further, though osteoarthritis was traditionally thought to be caused from “wear and tear”, recent advances in technology now point to a multifactorial pathogenesis including genetic predisposition, epigenetics, and sex, as well as lifestyle factors such as diet, physical activity and work-related habits [31]. Particularly, diet and obesity are significant players in the development of osteoarthritis risk factors because of their role in the pathogenesis of metabolic syndrome, which includes central obesity, diabetes, high blood pressure, and hyperlipidemia, and have been shown to be independently associated with the onset and progression of osteoarthritis [32].

#### 3.1.3. Ligamentum Flavum Thickening

As previously mentioned, both disc bulging and arthritic changes can contribute to lumbar spinal stenosis; however, ligamentum flavum (LF) hypertrophy is believed to be the main contributor [33,34]. The pathophysiology of LF hypertrophy remains unclear. Prior research has demonstrated that tissue thickening may be due to buckling [35]. However, other studies have concluded that thickening may occur mainly in the extracellular matrix indicating hypertrophy is to blame. Age, disc degeneration, and hemodialysis have been indicated as possible risk factors for LF thickening [35,36,37,38,39,40,41].

#### 3.1.4. Other Causes

Other lumbar spinal pathologies include spondylolisthesis, space occupying lesions, and infectious processes. Spondylolisthesis is the anterior displacement of a vertebrae and is associated with a bilateral defect of the pars interarticularis. Severity is based on the degree of displacement, and the need for surgical intervention is guided by symptom severity and the concern for instability [42]. Lesions such as cysts, masses, or abscesses can lead to the development of symptoms secondary to spinal cord compression. If symptomatic, compression of the spinal cord is a surgical emergency requiring immediate intervention to prevent permanent deficits. Infectious processes in the spine such as spondylodiscitis and spinal osteomyelitis also lead to various neurological sequalae. Often caused by the spread of infection from adjacent soft tissues or directly from spinal procedures, osteomyelitis/discitis can be a challenge to diagnose and adequately treat [43].

**Figure 4 biomedicines-11-00756-f004:**
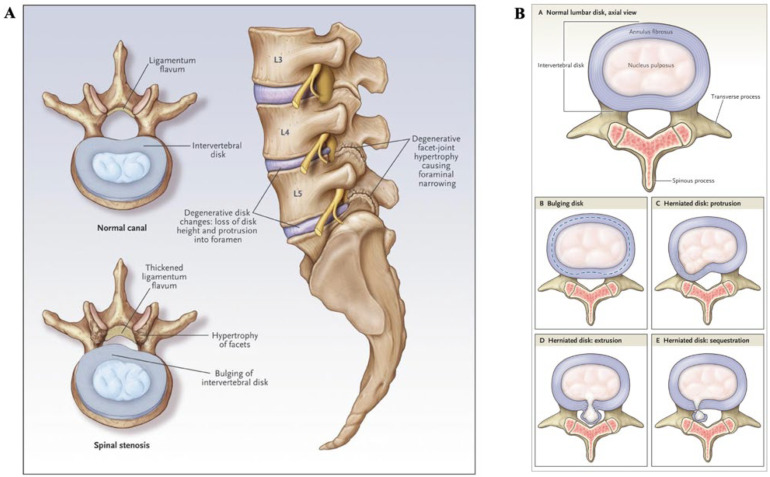
Characteristics of Lumbar Spinal Stenosis and Herniated Disks. (**A**) Axial views of the lumbar spine showing both normal anatomy and spinal stenosis. Depiction demonstrates intervertebral disk bulging, a thickened ligamentum flavum, and hypertrophied fact joints. Additionally, on sagittal view, there is loss of disk height and protrusion, and facet-joint osteoarthritis [44]. (**B**) Panel A demonstrates a non-displaced lumbar intervertebral disk. Panel B shows a bulging disk. Panel C demonstrates disk protrusion. Panel D demonstrates disk extrusion. Panel E demonstrates disk sequestration [45]. Adapted from Deyo et al. and Katz et al. [44,45].

### 3.2. Lumbosacral Plexopathy

#### 3.2.1. Trauma

Due to the LSP’s anatomic location deep within the pelvis, it is relatively shielded from direct injury. Because of this, traumatic plexopathies most likely result from penetration injuries such as a gunshot or puncture wound [46,47]. Typically, traumatic LS plexopathies are commonly associated with pelvic or hip fractures, however, the incidence of LSP complication in patients treated for these types of fractures is low (0.7%) [47]. Contrastingly, in cases involving sacral fractures or sacroiliac dislocations, the frequency is significantly higher (2.03%) due to the proximity of the plexus to both the sacral bone and its articulating sacroiliac joint [47].

#### 3.2.2. Metabolic and Inflammatory Causes

The LSP may be involved alone or along with nearby roots and nerves in immune-mediated or inflammatory disorders [48]. Although lumbosacral radiculoplexus neuropathy (LSPRN) most frequently occurs in patients with diabetes, it can also be found in non-diabetics. In a recent study, incidence of LSRPN was 4.16/100,000 overall. Further, diabetics had an incidence of 2.79/100,000 while non-diabetics had an incidence of 1.27/100,000 [49]. Amyloidosis and sarcoidosis can also cause lumbosacral plexopathies secondary to local inflammatory changes [50].

#### 3.2.3. Neoplastic

According to recent literature, for LS plexopathies caused by neoplasms, the L4-S1 segment is affected in more than 50% of cases, the L1–L4 in 31% of cases, and pan-plexopathy in approximately 10% of cases [51]. Plexopathies were found to occur within the first year after cancer diagnosis in one-third of cases and be the presenting symptom in 15% of cases [52,53]. Pain is the most common initial symptom followed by sensory or motor problems beginning within weeks to months [51].

#### 3.2.4. Other Causes

Other causes of lumbosacral plexopathies include chronic infections such as tuberculosis, fungal infections, lyme, HIV/AIDS, and herpes zoster [5]. Additionally, because of the proximity of the psoas major and minor muscle groups to the plexus, psoas abscesses can lead to various neurologic symptoms. Radiation can also lead to plexopathies and can present without pain and occur bilaterally even years after radiation [54]. Vascular causes include femoral catheterization and ischemic insults from direct compression caused by arterial pseudoaneurysms, aortic dissections, and retroperitoneal hematomas. Postoperative plexopathies can result from direct trauma or prolonged retraction.

## 4. Neurologic Consultation

With a foundational understanding of the etiologies of lumbar spinal pathology and LS plexopathies, we can better elucidate differences in clinical presentations and some of the current diagnostic modalities that serve to better characterize these pathologies.

### 4.1. Clinical Presentationand Physical Examination

#### 4.1.1. Distinctions in Symptomatic Presentation

In the initial evaluation of a patient presenting with symptoms of LBP, it is pertinent to evaluate differential etiologies and—for the purpose of this discussion—distinguish between lumbar spinal pathology and LS plexopathy.

Lumbar spinal pathologies typically present with the main complaint of chronic back pain. Important differences in a patient’s symptomatic presentation are relevant to note as these allow us to distinguish within specific pathological processes. For example, in lumbar spinal stenosis (LSS), patients complain of neurogenic claudication and pain exacerbated by prolonged ambulation, standing, and lumbar extension—which is relieved by forward flexion and rest [55]. While LSS presents with bilateral pain, sciatica tends to be unilateral radicular pain to the ipsilateral lower extremity [56]. Additionally, the pain is typically “burning” in nature with occasional associated parathesia [56]. Further, lumbar disc herniation can also present with pain following an acute event or spinal movement and, thus, may be indicative of instability or degenerative fracture of the pars at L5 [57].

Given the diverse etiologies of LS plexopathy, this pathology can present variably in the clinical space and can prove difficult to diagnose for said reason. Nonetheless, it is important to note that the symptoms of LS plexopathies tend to reflect the level of anatomical involvement of the plexus and the temporality of the injury. The most common presenting symptoms of LS plexopathy include LBP with unilateral radiation and possible association with positionality [5]. In contrast to LSS, patients typically complain of bilateral leg pain [58]. Furthermore, specific etiologies of LS plexopathies may be associated with more unique symptoms. This exceptionality in presentation is showcased when comparing diabetic LS plexopathy to LS plexopathy secondary to radiotherapy, in which one classically presents with proximal thigh pain and the other is often unremarkable for pain [59]. Lastly, symptoms of muscle weakness and atrophy often occur in more severe cases of disease [60].

#### 4.1.2. Physical Examination

Pertinent physical examination findings in LS plexopathy include a positive straight leg raise test, as well as asymmetric lower limb muscle weakness with asymmetrically absent or reduced deep tendon reflexes [5]. Whereas in some lumbar spinal pathologies such as LSS, as little as 10% of patients demonstrate positive straight leg testing [61]. Similarly, the straight leg raise test has variable specificity for diagnosis of sciatica but may be a more useful indication of lumbar disc herniation [56]. Interestingly, lumbar disc herniations can present with a contralateral positive straight leg test as a supplement to a positive ipsilateral straight leg test, increasing the sensitivity from 84% to 96% [62].

Another important component of the physical examination includes reflex testing as changes to the deep tendon reflexes can be present in both LS plexopathies and lumbar spine pathologies. Specifically, lumbar plexopathy can present with knee jerk reflex abnormalities, while sacral plexopathy can present with ankle jerk reflex abnormalities [5]. Like lumbar plexopathies, an absent or diminished patellar tendon reflex, also known as the Westphal Sign, is commonly found in lumbar spine pathology such as sciatica [63].

Additionally, thorough evaluation of the patient’s gait and functionality can provide further insight into pathologic differentiation [57]. The aforementioned holds true because while LS plexopathy can present with more of a dermatomal pattern of symptoms, some lumbar spine pathologies, such as lumbar HNP, must be examined for more mechanical changes affecting function. Additionally, changes will be seen in ankle dorsiflexion or plantarflexion depending on the positionality of an L5/S1 disc herniation laterally or centrally into the neural foramen, further elucidating any functional changes [57].

Further, demonstration of sensory loss and/or weakness upon physical examination also allows us to distinguish between LS plexopathy and lumbar spine pathologies. In LS plexopathy, sacral involvement may present with sensory loss of the medial thigh, anterior thigh, dorsum of the foot, and perineum [5]. More specifically, the sensory and motor loss seen in femoral and sciatic neuropathies—disorders of two major components of the LSP—are crucial to note as they can present similarly to lumbar spine pathologies, albeit there are some notable distinctions upon physical examination. Typically, patients with femoral neuropathy present with sensory loss of the anterior and medial leg [64]. Additionally, there may be weakness involving the quadriceps and iliopsoas muscles on examination [64]. In correlation with these physical examination findings, patients may also complain of difficulty with stairs, frequent falling secondary to knee weakness, and acute pain in the groin, back and lower abdomen [65]. The etiology of this femoral nerve injury is varied and can occur secondary to retroperitoneal hematomas during abdominal surgery, or after total hip arthroplasties, to name a few [65,66,67]. Moreover, sciatic neuropathies present with more robust foot changes on physical examination, namely foot drop pain, foot inversion, and toe flexion, as well as sensory loss of the upper third of the lateral leg [68]. In contrast, in LSS for example, sensory losses on physical examination in addition to numbness and tingling usually involve the entire lower extremity, rather than following a dermatomal distribution that is characteristic of LS plexopathy [55]. While correlation with symptoms and history is important, the physical examination findings associated with femoral and sciatic neuropathies provide adequate insight to avoid the inaccurate diagnosis of a lumbar spine pathology.

### 4.2. Diagnostic Evaluation

#### 4.2.1. Radiological Imaging

Imaging for both lumbar spine pathology and LS plexopathy typically involves an MRI with gadolinium contrast, and a computed tomography (CT) scan as a second line option when MRI is contraindicated [69]. In LSS, CT or MRI may demonstrate a diminished intraspinal canal area and anteroposterior spinal canal diameters to confirm diagnosis [44]. Whereas in some lumbar spinal pathologies, such as sciatica, imaging offers little clinical utility, as it is primarily a clinical diagnosis [56]. While CT and MRI offer more insight for some lumbar spine pathologies that may be associated with surgical pathology or structural deformities, LS plexopathy requires more nuanced imaging modalities [70,71]. MR neurography is especially useful for patients presenting with symptoms reflecting lumbosacral plexus or sciatic nerve involvement to better examine the lumbosacral plexus or sciatic nerve [72]. Extraforaminal lesions can better be revealed on MR neurography compared to traditional MR imaging [72]. Findings using this technique can include: fibrous and muscular entrapment, vascular compression, posttraumatic lesions, neuropathy secondary to ischemia, neoplastic and granulomatous infiltration, neural sheath tumors, scar tissue secondary to radiation, and neuropathy that is hypertrophic in nature [72]. Additionally, Gupta et al. noted the presence of a “trident sign” on contrast-enhanced CT and MRI of the lumbosacral region. This “trident sign” is associated with perineural cancer dissemination throughout the lumbosacral plexus, with diffuse thickening of the nerve plexus, and is predictive of adverse survival rates [73].

#### 4.2.2. Electroneurography and Electromyography Studies

Sensory conduction studies are especially useful in distinguishing these two pathologies. A key point in distinguishing LS plexopathy and lumbar spine pathology is the effect on sensory nerve action potentials. If this potential is not reduced in sensory nerve conduction studies, this implies a preganglionic process such as a radiculopathy [74]. Whereas, if the sensory nerve action potential is reduced, it is a postganglionic process, which would imply a plexopathy or mononeuropathy [74]. Motor nerve conduction to diagnose pure LS plexopathy should be focused to the femoral nerve [74]. Peroneal and tibial motor studies may also assess axonal loss in the lumbosacral plexus, however, the usefulness in elderly patients is unlikely given diminished or absent responses in this population [74].

Electromyography (EMG) is also helpful as part of the neurological evaluation to differentiate LS plexopathy and lumbar spinal pathology. Patients with LS plexopathy with referral for EMG studies are typically type 2 diabetes mellitus patients with weight loss, and anterior and posterior thigh and buttock pain [75]. LS plexopathy presents with electrophysiologic abnormalities in the distribution of, at minimum, 2 different peripheral nerves in a minimum of 2 different nerve root distributions [74]. Determining the extent of abnormalities in LS plexopathy requires that needle EMG examination cover L2-S1 innervated muscles and muscles innervated by the same root but not the same peripheral nerves [74]. Pure lumbosacral plexopathy can be further identified when the paraspinals are not affected on needle examination [76]. In contrast to LSS, EMG examination is often normal [77].

In sum, notable differences in clinical presentation and diagnostic evaluation can be used to offer patients accurate diagnosis and subsequently more tailored treatment for their respective disease pathology. Chiefly, these interventions include conservative and surgical approaches, which will be highlighted in the next section.

## 5. Treatment and Management

The overarching aim of management of pathologies related to the lumbar spine and lumbosacral plexus is to primarily mitigate the progression of the disease, restore functional status of the lower limbs, and preserve the quality of life for the patient [4,60,78,79]. As mentioned above, a shared characteristic between both classes of pathologies is the experience of intense pain, thus highlighting the necessity of alleviatory management care.

### 5.1. Conservative Approaches

As generally common for spinal pathologies, symptomatic management begins with conservative therapies [4,78,80]. For both lumbar spine pathologies and LS plexopathies, these therapies mainly consist of physical therapy and medications focused on providing pain relief. Physical therapy techniques consist of exercises reinforcing core strength, improving lumbar musculature flexibility, and correcting posture [81]. Limited evidence supports the effectiveness of physical therapy regimens in terms of restoring functionality and alleviating pain [82] and is so far confined to low power studies with small samples sizes [83,84]. In addition to the lack of empirical support for alleviatory physical therapy alone, studies have shown that physical therapy does not consistently provide adequate pain relief and functional strengthening following invasive decompressive procedures [85]. Alongside physical therapy, medications such as analgesics—NSAIDs, opioids, and gabapentin—are prescribed to further alleviate pain symptoms and restore quality of life, although, again, are met with variable effectiveness contingent on the initial intensity of pain, comorbidities, and patient age [86,87,88]. Lastly, patients with lumbar spine pathologies can undergo selective nerve root injections and translaminar epidural steroid injections to locally deliver long-lasting steroids to irritated spinal nerves. These procedures are typically more favorable in outcome compared to the alternative conservative approaches to pain management, with approximately 60–75% of patients reporting a significant reduction in pain [89].

### 5.2. Surgical Approaches

Surgical intervention is deemed necessary once it has been sufficiently determined that conservative therapies fail to yield substantial benefits to the patient. As with any form of surgery, it is of the utmost importance to consider potential complications, including repeat post-operative endotracheal intubation, cardiopulmonary resuscitation, pulmonary embolism, and many more. Furthermore, these complications can increase in frequency with various commonly encountered comorbidities such as advanced age, obesity, or diabetes [90].

#### 5.2.1. Laminectomy for Lumbar Spinal Stenosis

Typically, the primary goal of surgical intervention for patients with lumbar spinal stenosis is to improve lower limb functionality, and it is often decided upon following a three-to-six-month period of conservative treatments met with failure to reduce pain and further neurological impairment [91]. The most common surgical procedures indicated for lumbar spinal stenosis are laminectomies [92]. Decompressive lumbar laminectomy procedures entail the excision of the lamina and spinous process at the level at which nerve compression occurs. A successful procedure with resection of the lamina and excessive facet joints results in the respective decompression of the central canal and neural foramina, thus relieving pressure and preventing further nerve impingement [93].

A retrospective analysis performed by Bydon and colleagues demonstrated significant positive outcomes with regards to neurogenic claudication and motor function in patients following lumbar laminectomy, highlighting its relative effectiveness compared to the inconsistent outcomes observed with conservative approaches [94]. In fact, the study emphasizes that the percentage of patients who reported initial lower back pain decreased from 57.40% to 25.40% (*p* < 0.001) following the procedure [94]. Despite its effectiveness for the treatment of lumbar spinal stenosis, the laminectomy procedure is not without complications. A primary concern for laminectomy for lumbar spinal stenosis involves the comorbidity of preoperative spondylolisthesis, as well as iatrogenic spondylolisthesis following the procedure [95]. A systematic review including 2496 patients determined that 1.8% patients that underwent decompressive lumbar laminectomy developed instability that warranted a reoperation [95]. This phenomenon was observed more frequently with patients with pre-existing spondylolisthesis prior to operation [95]. Currently, there is thought that with minimally invasive techniques, the risk of instability and therefore reoperation deceases, although it may achieve less than desirable decompression. Additionally, the field still requires larger prospective studies to verify the effectiveness of these minimally invasive techniques [95]. Otherwise, instability could potentially be avoided through the combination of lumbar laminectomy with spinal fusion [96,97].

#### 5.2.2. Discectomy for Lumbar Herniated Nucleus Pulposus

For patients that fail to experience symptomatic benefits from conservative treatments, lumbar discectomy is typically indicated [98]. These procedures are described by excision of the herniated disk following an opening of the spine and dissection of nearby nerve roots if necessary [99]. The incremental excision process does not typically remove the entire disk, as it is terminated after two to three failed attempts to remove any additional material with the forceps. There are also microendoscopic alternatives to this approach (Figure 5B) [45,99]. However, studies fail to demonstrate a difference in postoperative pain outcomes between the two versions of the procedure [99]. Compared to conservative approaches, lumbar discectomies demonstrate better outcomes regarding pain. Furthermore, the location of the discectomy has a remarkable effect on the favorability of the patient outcomes, as it was demonstrated that upper-level lumbar herniation repairs are slightly better in outcome than lower-level lumbar repairs [100]. Most patients experience an alleviation of back and leg pain following procedure, yet one study determined that 28% of patients still experience pain symptoms [101]. Furthermore, there is risk for recurrent disc herniation following procedure, warranting reoperation (7.3% of patients) [101].

#### 5.2.3. Surgical Intervention Strategies for Lumbosacral Plexopathy

In direct contrast to lumbar spine injuries, LS plexopathy lacks clearly established surgical guidelines, let alone decompressive techniques, due to its inherent locational and functional complexity [60]. Further complicating matters, LS plexopathies can secondarily arise from a variety of primary conditions, such as diabetes mellitus, neoplastic developments, or complications in radiotherapy [5]. Surgical intervention for LS plexopathies is typically very involved on the part of the surgeon, requiring extensive navigation such as extraperitoneal lumbotomy, transperitoneal xifopubical incision, or posterior laminectomy, to reach the targeted nervous structure depending on the level location of the structure [102].

Following these procedures to expose the targeted nerve, surgeons typically opt for neurolysis to mitigate pain symptoms related to the pathology of that peripheral nerve, which often is the femoral, obturator, and the sciatic nerve when considering LS plexopathies [103]. As discussed above, pathologies related to these nerves often result in deambulation and the radiation of burning pain [7,58]. As such, the overall goal of these surgeries is to alleviate these symptoms directly related to those nerves. Neurolysis entails the circumferential dissection of the targeted nerve, with the goal of removing any physical obstruction (i.e., scar tissue) that may impact the route and thus transmission of the nerve [104,105]. This peripheral approach to treatment, when targeting the femoral nerve in patients experiencing LS plexopathy, has been found to attain positive results with various patients, including a reduction in severe pain and restored motor function to the gluteal muscles [102,106]. Furthermore, in studies following patients with plexopathy affecting on the obturator nerve, and thus experiencing limited hip adduction, it was determined that neurolysis on the obturator nerve improved hip adduction from Medial Research Council (MRC) grade 2 to MRC grade 5 along with a reduction in pain [103,107].

Further, repair strategies used in LS plexopathies with extensive neuronal damage include procedural nerve grafts to restore function to affected peripheral nerves (Figure 5A) [60,103]. Nerve grafts can either entail an end-to-end nerve suture or suturing an autologous donor nerve to replace the damaged portion of the recipient nerve [108,109]. Various case reports highlight the effectiveness of this strategy, namely restoring the functionality of the femoral, obturator, and sciatic nerves, as well as for multilevel injuries involving nerve roots [102,103,107,110,111]. Across these various procedures, it was reported that related motor and sensory function was successfully restored, including hip adduction, hip flexion, and knee extension [102,103,107,110,111].

## 6. Emerging Treatment Options

As aforementioned, treatment for lumbar spine pathologies and LS plexopathies include both conservative and surgical approaches. More recently, emerging treatments have taken advantage of new approaches or modified previous approaches, specifically with the aim of nerve regeneration.

### 6.1. Novel Conservative and Surgical Techniques

#### 6.1.1. Approaches for Common Lumbar Spinal Pathologies

Regarding sciatica, the most common pathology of the lumbar region, a nerve root foramen opening protocol was recently used as conservative treatment [112]. This approach involves placing patients in a side lying position to invoke opening of the foramen. Results have showed further alleviation of pain and reduction of opioid consumption with this approach [112]. Further, this approach is significantly simpler than previous therapies and allows the luxury for patients to perform this manually at home, reducing costs of care. Although further research is needed to prove the effectiveness of this regimen, it is a step in a new direction for sciatica conservative care.

Sciatica can often be caused by lumbar spondylisthesis (LS), which is defined by the “slippage” of a vertebral body [113]. Recently, several clinical trials are in place to find the most optimal and effective treatment for LS. The Spine Patient Outcomes Research Trial (SPORT) has shown the greater effectiveness of surgery in relation to nonoperative care for LS [114]. However, the lack of focus on the superiority of laminectomy plus fusion in relation to just laminectomy in treatment of LS has led to the Spinal Laminectomy versus Instrumented Pedicle Screw (SLIP) trial [96]. Results show that utilization of lumbar spinal fusion in combination with laminectomy showed clinically meaningful improvement in quality of life of patient’s post-surgery [96].

Additionally, LSS also contributes to the pathogenesis of lumbar sciatica [115]. Typically, patients with LSS have a variety of conservative options including pain medications and physical therapy. Surgical intervention for LSS is common in cases when patients begin to see neurological deterioration and symptoms [115,116]. The standard of care has been decompression surgery, requiring significant manipulation of spinal muscles [116]. This approach has been seen to increase the risk of postoperative complications [117]. Emerging treatments for lumbar stenosis have adopted a minimally invasive surgical approach using endoscopy, with the aim of decreasing surgical trauma and maximizing preservation of the spinal structure. Recently, unilateral biportal endoscopic spinal surgery (UBESS) has been a promising new approach [118]. The design includes two portals in comparison to traditional spinal endoscopy with one portal [118]. This allows UBESS a wider range of movement and accounts for the shortcomings of traditional endoscopy of a lack of proper visualization. More recent studies have shown the effectiveness of UBESS for LSS, but further research is needed to show further efficacy.

#### 6.1.2. Approaches for Lumbosacral Plexopathies

As previously mentioned, LS plexopathies have historically not had a definitive standard surgical guideline due to their inherent complexity [28]. Surgical options such as extraperitoneal lumbotomy, and posterior laminectomy tend to be the approach, followed by neurolysis [71]. In cases of extensive neuronal damage, opting for procedural nerve grafts is also feasible to restore nerve function [28,72]. More recently, the pararectus approach has been utilized for both anterior exposure and neurolysis of lumbar nerve roots 4/5 [119]. Originally, this technique used in acetabular fracture repairs, but has now has utility in wide arrange of procedures including resection of musculoskeletal tumors [120,121]. This intrapelvic technique involves specific positioning of the patient in a supine position that puts the groin, umbilicus, and anterior superior iliac spine (ASIS) in a triangular arrangement [119]. In comparison to the initial use of the pararectus approach, the neurolysis for LSP involves an incision being roughly 4–5 cm closer towards the cranium [119]. After superficial and deep dissection to visualize the anterior abdominal wall and tranversalis fascia, respectively, surgeons work through the retroperitoneal space bluntly [119]. Following the iliac vessels gently and reaching the L4 and L5 nerve roots along with the obturator nerve allows for proper neurolysis [119]. Although scarce research is available regarding this approach for LS plexopathy, the exposure of the iliac vessels for effective visualization of the LSP combined with and lack of osteotomy demonstrates its procedural advantages. However, further research is needed to smooth some limitations such as the difficulty in treating obese patients and those with previous scarring [119]. Nonetheless, results in cases thus far have shown to be effective with good outcomes [119].

Less invasive approaches for malignant lumbosacral plexopathy have recently been explored as well. Specifically, in the rare malignancy of the LSP, radiation therapy has proven to provide a route for significant pain relief in patients with lumbosacral syndrome and should be investigated further as a viable treatment option [51,122]. More recently, MR neurography (MRN) has also proven valuable in visualization of the peripheral nerves to allow for easier nerve tracking [123]. Due to this radiotherapy potentially causing neurotoxicity at higher doses, accurate contouring becomes an important aspect of treatment [124]. MRN’s visualization in combination with CT allows for personalization of contouring by the physician and less need for anatomical knowledge [123]. This is especially made possible through utilizing a 3D MRN sequence such as Lr_NerveVIEW, which has advantageous features that improve LSP nerve contouring [123]. Further, with the likelihood of advanced imaging techniques on the horizon, it only makes this approach more intriguing. However, more research is needed in a larger sample size before this is implemented on a greater scale.

### 6.2. Nerve Regeneration Strategies

Further, focus has also been on emerging treatment options for nerve regeneration and the coordination of these procedures with the standard surgical options previously mentioned. Although peripheral nerves can regenerate, this is a slow process and patients with neurotmesis or axonotmesis typically experience an impaired quality of life [125]. Conservative approaches to nerve regeneration have primarily included physical rehabilitation using various exercises [125]. This approach has been the standard non-invasive strategy and has been seen to improve peripheral nerve injury (PNI) recovery [126,127]. Recent advances in stem cell therapy have shown the potential of cell transplantation therapy for nerve regeneration. Strategically reconstructing lesion sites with stem cell grafts, which will later differentiate into neurons, has been a valuable approach to nerve regeneration [128]. Utilization of this strategy entails the transplantation of embryonic and pluripotent neuronal stem/progenitor cells (NSCs/NPCs) into lesion sites, where they will eventually develop into self-integrating grafts within the tissue [128,129]. The connection of the implanted neurons with the hosts circuitry provides a route for new sensory connections. Further, axons have been shown to extend into the graft and further synapse with the graft neurons allowing for the formation of a new circuit [130]. Although, this approach is promising and exciting, many axons are at the superficial surface of the graft (1 mm) and will not significantly integrate into the graft. Further research exploring adverse effects and consistency of this strategy must be done prior to having a clinical impact. More recently, mesenchymal stem cells (MSC) have been identified as potentially valuable for nerve regeneration and lack the ethical concern component of their aforementioned counterparts [131]. Particularly, research into MSCs ability to integrate into host lesions has shown superior migration ability due to various factors [132,133]. The ability for MSCs to release neuroprotective factors and control glial scarring makes them particularly attractive, but further research must be done with the aim of translating these findings to clinical practice.

Finally, the most recent development surrounding novel nerve regeneration therapeutics lies in the electrical stimulation of transplanted neural stem cells and the surrounding musculature via “nerve conduits”. These conduits, or “nerve guidance channels”, are complex biomaterials that offer injured peripheral nerves a scaffolding to promote guided axonal growth through physical, chemical, and electrical means [134]. Particularly, newer conduit materials such as polypyrrole (PPy) have come under extensive investigation for their unique conductive properties under physiologic conditions and their excellent biocompatibility in vivo [135]. These capacities as mentioned above of PPy are of the utmost importance in novel peripheral nerve regeneration because it has been found that the direct electrical stimulation of peripheral nerves leads to an increased release of neurotropic chemicals such as brain-derived neurotrophic factor (BDNF)—an essential small molecule involved in axonal regeneration and re-myelination [136]. Furthermore, BDNF has modulatory effects on tropomyosin receptor kinase (Trk), whose pathway activation has multiple downstream effects involved in stimulating the “growth cone” during nerve repair processes [137]. Cumulatively, by manipulating the physiological microenvironment and providing a robust biocompatible framework using electrical stimulation and nerve conduits, this is a step towards amplifying the therapeutic effect of stem cell-based treatments on neuronal regeneration [138].

## 7. Conclusions

In this review, we introduced a common ailment—lower back pain and its classically associated symptoms of radiating lower limb pain and muscle weakness—and explored two groups of peripheral neurologic disorders that can lead to said symptomatic manifestations. The first of these disorders, lumbar spine pathology, was found to be comprised of various disease states such as HNP, osteoarthritis, and ligamentum flavum thickening, all with a common etiology of lumbar spine stenosis. The rarer of the two, LS plexopathies, were described as less mechanically involved and were principally caused by metabolic and inflammatory processes. Clinically, since the presentation of lumbar spine pathology and LS plexopathies tend to be indistinguishable upon superficial assessment, we highlighted distinctive symptomatic presentations such as the bilateral leg pain most often reported in patients with a lumbar spine pathology, in contrast to the unilateral radiation found in LS plexopathy patients. Of importance, we also identified key physical examination findings that underscored the differences in sensory and motor deficits, such as the involvement of the entire lower extremity versus following a dermatomal distribution, in lumbar spine pathology and LS plexopathy, respectively. In conjunction with the latter clinical forms of assessment, we described useful diagnostic testing such as contrast MRI and CT to grossly examine nervous structures, as well as detailed the more nuanced techniques of sensory and EMG studies to further elucidate specific involvements. Additionally, we discussed the utility of conservative and surgical approaches for the purpose of alleviatory care in the treatment of both pathologies, although the literature was much scarcer regarding surgical interventions in the treatment of LS plexopathies. Finally, we looked to the future of treatment for these pathologies in the context of novel nerve regeneration therapeutics and their integration into standard-of-care approaches via the employment of NSCs/NPCs, their mesenchymal counterparts, and their combined use with conductive nerve conduits.

## Figures and Tables

**Figure 1 biomedicines-11-00756-f001:**
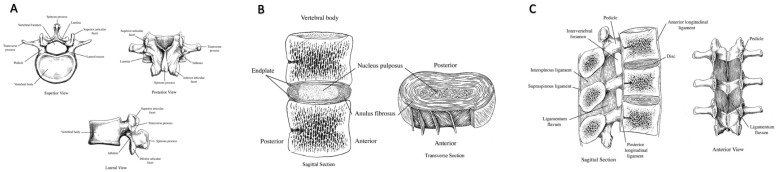
Anatomy of the Lumbar Spine. (**A**) Superior, posterior and lateral aspect of the lumbar vertebrae and the associated osseous structures. (**B**) The sagittal and transverse cross-sections of the lumbar intervertebral disc and its three component parts—the nucleus pulposus, annulus fibrosus, and endplate. (**C**) The sagittal and anterior aspect of the articulating lumbar vertebrae—note the intervertebral foramen formed between the pedicles of the neighboring vertebrae [9]. Adapted from Ebraheim et al. [9].

**Figure 2 biomedicines-11-00756-f002:**
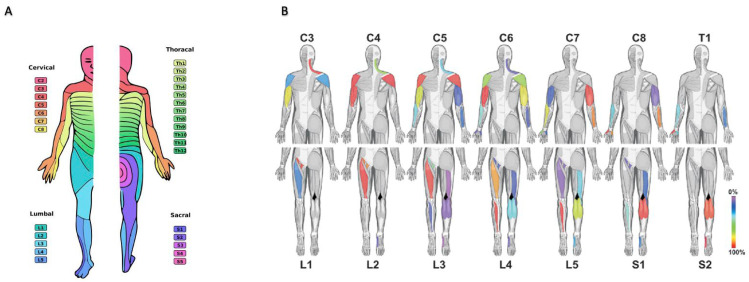
Overview of the Dermatomal and Myotomal Innervations of the Spinal Nerves. (**A**) Diagram of the respective dermatomal distributions for the 31 pairs of spinal nerves as they travel along their distinct paths from posterior to anterior [15]. (**B**) Myotomal distributions of the lumbosacral spinal nerves. The muscle groups are color-mapped to reflect the rate of response upon individual intraoperative nerve root stimulation [16]. Adapted from Whitman et al. and Schirmer et al. [15,16].

**Figure 3 biomedicines-11-00756-f003:**
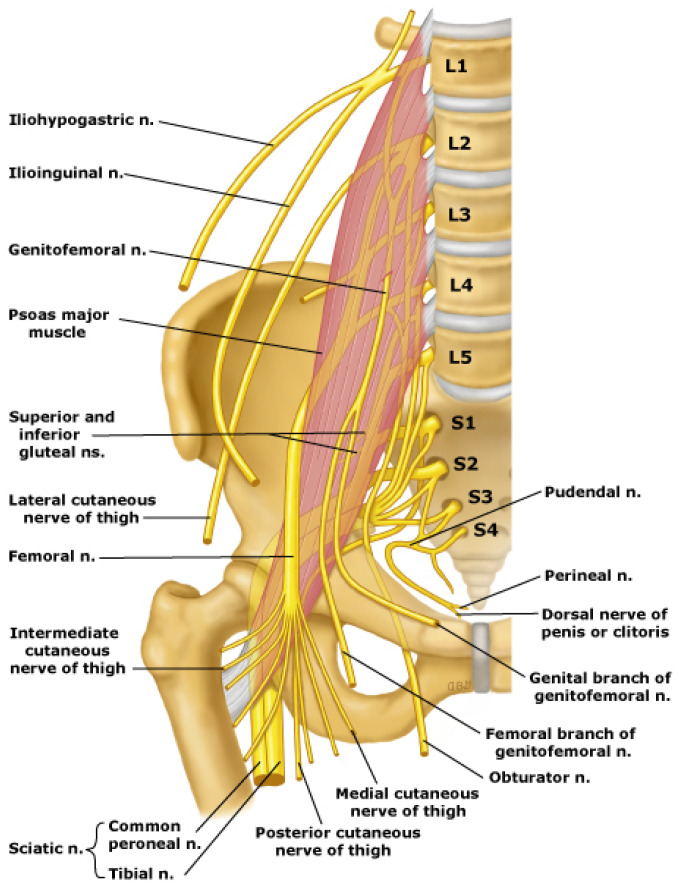
Anatomy of the Lumbosacral Plexus. Coronal view of the lumbosacral plexus and its branches [19].

**Figure 5 biomedicines-11-00756-f005:**
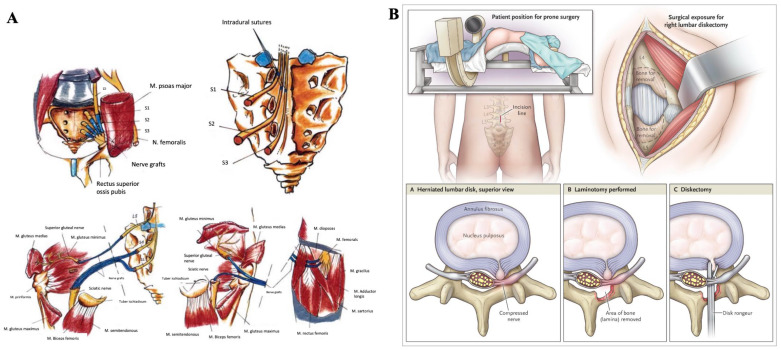
Repair strategies for lumbar spinal pathology and lumbosacral plexopathy. (**A**) Repair strategies used in lumbosacral plexopathy. Upper Left: Intrapelvic repair of the sacral plexus. Upper Right: Intraspinal repair of ruptured ventral roots with sutured nerve grafts. Lower Left: Intraspinal and extrapelvic reconstruction of the sacral plexus by means of nerve grafts to gluteal and sciatic nerves. Lower Right: Nerve transfers by means of nerve grafts from fascicles of femoral nerve to gluteal and sciatic nerves [60]. (**B**) Technique of microdiscectomy is shown. Panel A shows a posterolateral disk herniation. Panel B shows a small incision made with a surgical microscope and a small laminotomy. Panel C shows a discectomy being performed [45]. Adapted from Deyo et al. and Lang et al. [45,60].

**Table 1 biomedicines-11-00756-t001:** Nerves of the lumbosacral plexus and their respective muscular and sensory innervations [20].

	Nerve	Muscles	Sensory Distribution
Lumbar Plexus	Iliohypogastric (L1-2)	-	Inferior abdominal wall
	Ilioinguinal (L1-2)	-	Medial Groin
	Genitofemoral (L1-2)	-	-
	Lateral femoral cutaneous (L3-4)	-	Anterolateral thigh
	Obturator (L2,3,4)	Adductor long Adductor magnus Gracilis	-
	Femoral (L2,3,4) Saphenous (L2,3,4)	Quadriceps -	- Medial leg and foot
Sacral Plexus	Sup. Gluteal (L4-5)	Gluteus medius Tensor fascia lata	-
	Inf. Gluteal (L4-S1)	Gluteus maximus	-
	Sciatic (L4-S2)	Anterior tibialis Peroneus longus Gastrocnemius Soleus Foot muscles	Foot Lateral leg
	Pudendal (S2,3,4)	External anal sphincter	Perineal

## Data Availability

No new data were created or analyzed in this study. Data sharing is not applicable to this article.

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
