# Peer review of "Differentiating Lumbar Spinal Etiology from Peripheral Plexopathies"

_biomedicines, 2023, doi:10.3390/biomedicines11030756_

Round 1

Reviewer 1 Report

The review deals with the urgent issue of differential diagnosis between lumbar plexopathy and sciatica as the diseases commonly classified as related to low back pain symptoms. The problem is important and expected by the clinicians. Presentation of anatomical origins and clinical presentation of the diagnosis of these two disorders are greatly appreciated.

The anatomical aspects are presented in an outstanding way, hope that anatomical drawings deriving from prominent handbooks are permitted by their creators. In fact, this article does not have weak points except for some difficult-to-understand phrases and editorial corrections.

Abstract

Lines 13-14. …”Since said pathologies present comparably in the clinical setting, hasn't proven easy to differentiate between the two peripheral neurologic disorders.”…

Sciatica is a consequence of the disc-root conflict, “peripheral neurologic disorders” suggest plexopathies, mono- or polyneuropathies. The sentence should be rewritten.

Lines 14-15, …” Resultingly, the inappropriate diagnosis of one pathology over the other can have consequential downstream effects on effective treatment and management.”… Hmm… “Treatment” and “management” in one sentence.

Lines 20-21 …”Lastly, this review will describe current preclinical and clinical progress towards the incorporation  of regenerative nerve therapy to further improve patient outcomes.”… you meant outcomes of the patient’s therapy?

Main content

Section 4.4.1. …”Comparatively, the symptomatic presentation of LS plexopathy is varied given its diverse etiologies.”… unclear needs rewriting.

Subsection title “4.2.2. Electroconductive and Sensory Studies”     should be changed to “Electroneurography and electromyography studies” considering its content.

The lack of the presentation among the electrodiagnostic procedure s “Motor evoked potentials” induced with the magnetic field oververtebrally is disappointing.

Subsection 6.2. Nerve Regeneration Strategies does not include the advances in electrotherapy of nerves and muscles . Their effectiveness in the treatment has been proven.

Editorial minor revisions.

General separation of Refs. in brackets from sentences throughout the manuscript is necessary.

Refs. list preparation is not in accordance with the MDPI style.

Delete in lines 620-621 …”For research articles with several authors, a short paragraph specifying their  individual contributions must be provided. The following statements should be used “…

Author Response

We greatly appreciate the comments provided by Reviewer 1 and 2. 

Abstract

1. Lines 13-14. …”Since said pathologies present comparably in the clinical setting, hasn't proven easy to differentiate between the two peripheral neurologic disorders.”…

Reworded the sentence for easier-to-understand phrasing.

2. Sciatica is a consequence of the disc-root conflict, “peripheral neurologic disorders” suggest plexopathies, mono- or polyneuropathies. The sentence should be rewritten.

Comment is previously addressed with edits made to line 13-14.

3. Lines 14-15, …” Resultingly, the inappropriate diagnosis of one pathology over the other can have consequential downstream effects on effective treatment and management.”… Hmm… “Treatment” and “management” in one sentence.

Removed redundancy of treatment and management and made sentence more clear to the reader.

4. Lines 20-21 …”Lastly, this review will describe current preclinical and clinical progress towards the incorporation  of regenerative nerve therapy to further improve patient outcomes.”… you meant outcomes of the patient’s therapy?

Reworded sentence to make it more clear to the readery.

Main content

5. Section 4.4.1. …”Comparatively, the symptomatic presentation of LS plexopathy is varied given its diverse etiologies.”… unclear needs rewriting.

Sentence rewritten for more clear understanding to the reader.

6. Subsection title “4.2.2. Electroconductive and Sensory Studies”     should be changed to “Electroneurography and electromyography studies” considering its content.

Accepted reviewer suggestion and changed title of heading.

7. The lack of the presentation among the electrodiagnostic procedure s “Motor evoked potentials” induced with the magnetic field oververtebrally is disappointing.

This section of the paper serves as a brief overview of electroneurography/electromyography studies and serves its purpose in our paper. Further writing in this section is outside of the scope of our paper.  

8. Subsection 6.2. Nerve Regeneration Strategies does not include the advances in electrotherapy of nerves and muscles . Their effectiveness in the treatment has been proven.

New paragraph highlighting novel regenerative strategies via electrical stimulation.

Editorial minor revisions

9. General separation of Refs. in brackets from sentences throughout the manuscript is necessary.

Separation of references homogenous throughout manuscript.

10. list preparation is not in accordance with the MDPI style.

Reference list prepared in accordance with ACS Style Guide, per MDPI’s Reference List and Citations.

11. Delete in lines 620-621 …”For research articles with several authors, a short paragraph specifying their  individual contributions must be provided. The following statements should be used “…

Deleted these lines per reviewer suggestion. 

Reviewer 2 Report

The Manuscript: „ Differentiating Lumbar Spinal Etiology from Peripheral Plexopathies’’ by Marco Foreman and colleagues highlighted the clinical differentiation between lumbar spine pathology and lumbosacral plexopathy by delineating the clinical assessment of these pathologies, aiming to bolster an appropriate diagnosis and treatment of these pathologies. The review amply offers a current understanding of themes on umbar Spinal Etiology and Peripheral Plexopathies. After going through the manuscript, I have a couple of comments for the author:

1.     Sometimes plexopathies are misunderstood as radiculopathies. I would suggest to include a sentence highlighting the differences between plexopathies and radiculopathies.

2.     Please mention how plexopathies are diagnosed and what are the main challenges in diagnosis of plexopathies.

Author Response

We greatly appreciate the comments provided by Reviewer 1 and 2.

1. Sometimes plexopathies are misunderstood as radiculopathies. I would suggest to include a sentence highlighting the differences between plexopathies and radiculopathies.

Paragraph in the beginning of the introduction should highlight this difference now.

2. Please mention how plexopathies are diagnosed and what are the main challenges in diagnosis of plexopathies.

In section 4 we added a statement mentioning why plexopathies are challenging to diagnose as well as discussed in detail some of the specific diagnostic tools that are useful for their diagnosis. For example, we discuss the use of reflex testing, patterns of sensory loss and/or weakness, as well as EMG and electroneurography studies that are specific to LS plexopathy diagnosis.